# Chromatophoromas in Reptiles

**DOI:** 10.3390/vetsci9030115

**Published:** 2022-03-04

**Authors:** Colleen F. Monahan, Michael M. Garner, Matti Kiupel

**Affiliations:** 1New Hampshire Veterinary Diagnostic Laboratory, University of New Hampshire, Durham, NH 03824, USA; 2Northwest ZooPath, Monroe, WA 98272, USA; zoopath1@gmail.com; 3Veterinary Diagnostic Laboratory, Michigan State University, Lansing, MI 48910, USA; kiupel@msu.edu

**Keywords:** reptile, chromatophoroma, melanoma, melanophoroma, iridophoroma, xanthophoroma, neoplasia, lizard, snake, turtle

## Abstract

Chromatophoromas are neoplasms that arise from pigment cells of reptiles, amphibians, and fish. They include melanophoromas (melanomas), iridophoromas, and xanthophoromas. Most chromatophoromas develop spontaneously, but genetic and environmental factors may also play a role in their oncogenesis. The diagnosis is typically through histologic examination. Immunohistochemistry and electron microscopy can be helpful for diagnosing poorly differentiated and/or poorly pigmented neoplasms. Aggressive surgical excision is the current treatment of choice. This review describes the clinical presentation, gross appearance, diagnostic applications, clinical behavior, and treatment of chromatophoromas in reptiles.

## 1. Background

The color of the skin of animals is determined by the amount of pigment within the epidermis and/or dermis that is produced by pigment cells. In mammals and birds, melanocytes are the sole type of pigment cell [1,2], whereas reptiles, amphibians, and fish possess a variety of pigment cells collectively termed chromatophores [2,3]. Chromatophores include melanophores (or melanocytes), iridiophores, leukophores, cyanophores, xanthophores, and erythrophores [1,2,3]. The latter two are often described collectively as xanthophores [4]. Melanophores, iridophores, and xanthophores have been described in reptiles [1,2,3]. Chromatophores in reptiles are typically localized in the dermis, which contrasts with mammalian epidermal melanocytes. Few reptile species are reported to also possess epidermal melanocytes [5]. Chromatophores or chromatophore-like cells (not melanomacrophages) are also commonly encountered in the coelom, especially in the parietal coelomic membrane of lizards, and scattered through the lung and liver [6]. The different types of pigment cells account for the wide variety of coloration observed in reptilian species and the ability, in some, to undergo rapid color changes [2,3]. These color adaptions aid in camouflage, communication, and sexual selection in these species [4].

Similar to mammalian melanocytes, chromatophores share a common stem cell of neural crest origin [6]. This stem cell migrates to the dermis during embryonic development and then further differentiates into the various chromatophores distributed throughout the dermis [6]. During differentiation, the membrane bound pigment organelles develop from the endoplasmic reticulum or vesicles from the Golgi apparatus that differ based on the respective types of chromatophores [6].

Chromatophores are divided into light-absorbing, pigment-producing chromatophores (melanophores and xanthophores) and structural or light-reflecting chromatophores (iridophores) with different genes involved in the development and differentiation of these two functionally different classes of chromatophores [7]. Melanophores contain melanin-containing melanosomes. Melanin, of the tyrosine-derived class of pigments, absorbs most light and therefore appears black to brown [1,6]. Xanthophores contain pteridine-containing pterinosomes and/or carotenoid-filled lipid vesicles, which absorb wavelengths of light resulting in yellow to red coloration [1,3]. Iridophores contain purine or pteridine-containing platelets, which reflect light and manifest as blue coloration or iridescence [1,3]. Guanin is the most common purine constituent within iridophore platelets, but hypoxanthine, adenine, and uric acid are also components [6]. Variations in the arrangement, size, number, and conformation of these platelets scatter light at different angles, resulting in color variation [5].

The normal arrangement of the chromatophores within the dermis, termed the ‘dermal chromatophore unit’ allows for the diversity of colors and rapid color change seen in reptiles [3]. The latter is especially prominent in chameleons (*Chamaeleo* sp.). There is a thin layer of iridophores within the superficial dermis just below the basal lamina of the epidermis [3]. Underneath this layer is a layer of xanthophores, and directly beneath this layer are the melanophores [3]. The melanophores have dendritic processes that extend between and around the overlying chromatophores [3]. Through neural or hormonal influences, melanophores can move their intracytoplasmic melanosomes into these processes via microtubules and actin filaments [3,4]. When these melanosome pigments are within the processes that cover the iridophores, the light that reaches the light-reflective iridophores is obscured, which manifests as a skin color change [3,4]. When the xanthophore and melanophore pigments are interpreted together, the colors appear as darker pigmentation along the light spectrum [3,4].

All chromatophores have the potential to undergo neoplastic transformation and are described collectively as chromatophoromas, or more specifically as melanophoromas (or melanomas), iridophoromas, and xanthophoromas [8,9,10,11,12,13,14]. Mixed or combination chromatophoromas involving multiple chromatophore varieties within the same tumor also occur [8,9,12,13,14,15,16].

This review describes the clinical presentation, gross appearance, diagnostic applications, clinical behavior, and treatment of chromatophoromas in reptiles to elucidate commonalities and differences between groups, families, and species.

## 2. Chromatophoromas in Reptiles: Overview and Gross Appearance

In general, multiple varieties of chromatophoromas have been described in reptiles. Melanophoromas and iridophoromas are most common, with few reports of mixed chromatophoromas, xanthophoromas, and amelanotic or non-pigmented chromatophoromas [8,11,12,13,17,18]. Of the different reptilian orders, they are most frequently described in squamata with few reports in testudines (chelonia) and no reports in crocodilia or rhynchocephalia [9,11,12,13,14,19,20,21] Affected animals are almost always adults and no sex predilection is apparent [9,11,12,13,14]. Most reports are cutaneous chromatophoromas that develop as single or multiple masses at various sites, including the head, trunk, limbs (when applicable), and tail [9,12,13,22,23]. Chromatophoromas most often present as raised, pigmented masses, but more subtle alterations in scale pigmentation are less commonly described [9,12,13,14] (Figure 1a–e). Melanophoromas are often described as dark brown to black [8,9,12,13,24]. Iridophoromas are often described as white [8,9,25]. Xanthophoromas may be yellow, orange, or red [8,9,16,26]. Benign and malignant forms of each subset can occur, the latter sometimes with widespread metastases [8,9,11,12,13,14].

There are a few reports of primary oral chromatophoromas [8,9,17], a single report of a primary pulmonary melanophoroma in a beaded lizard (*Heloderma horridum exasperatum*) [27], a single report of a primary small intestinal chromatophoroma in a green tree python (*Chondropython viridis*) [28], and a single report of a primary iris melanoma in a gecko (unspecified) [11] (Figure 1f, different animal).

## 3. Etiology of Chromatophoroma in Reptiles

The vast majority of reptilian chromatophoromas seem to develop spontaneously and no specific etiology has been identified. However, few mechanisms of oncogenesis have been investigated in some species.

### 3.1. UV Radiation

Artificial ultraviolet (UV) radiation, such as in solariums or tanning beds, has been associated with an increased incidence of cutaneous melanoma in humans [29]. In most reptiles, exposure to adequate natural or artificial UV light is essential for the synthesis of vitamin D3 and calcium metabolism [30,31]. Therefore, given that many captive reptiles are exposed to artificial UV light daily, it has been speculated that UV radiation may contribute to the development of cutaneous chromatophoromas [8,14]. This theory is supported by the increased incidence of cutaneous chromatophoromas in sun-loving, day active reptiles, such as bearded dragons (*Pogona* sp) and the paucity of tumors in nocturnal reptiles [8,9,13]. This theory may be further supported by a series of wild-caught elapid snakes in Australia that developed cutaneous chromatophoromas on the lateral and dorsal body surfaces [24]. However, in one retrospective study of artificial UV light exposure in various reptiles with cutaneous chromatophoromas, neoplasms were reported in several animals that did not receive artificial UV light as part of their husbandry [8]. Interestingly, a review of the archives at Northwest ZooPath (private veterinary diagnostic laboratory, Monroe, WA, USA) identified 87% (52/60 where the anatomical location was noted, appendages were not included) of tumors occurred on the dorsum or lateral aspects of the body, while only 14% (8/60) occurred on the ventrum, periocular region, or in the eye. Primary visceral chromatophoromas were rare and included tumors arising from one of each of the following tissues/organs: lung, kidney, epicardium, and colon.

### 3.2. Genetics

A genetic predisposition to developing cutaneous and oral iridophoromas has been postulated in the “lemon frost” color variant of the leopard gecko (*Eublepharis macularius*) [25,32]. This population is highly inbred due to selective breeding for unique color variations, and has a high incidence of iridophoromas [25,32] (Figure 2a–d). Guo et al., reported that 80% of the homozygous leopard geckos (from a large captive breeding colony) that had the *lemon frost (lf)* allele developed iridophoromas between 6 months to 5 years of age [25]. They localized the mutation to a single locus containing a strong candidate gene, serine peptidase inhibitor, Kunitz type 1 (SPINT1) [25]. SPINT1 is a tumor suppressor gene that has been implicated in the development of human cutaneous melanoma and unregulated proliferation of epithelial cells in mice and zebrafish [33,34]. It was concluded that a defect in SPINT1 could be leading to excessive proliferation and neoplastic transformation of iridophores within this inbred population of leopard geckos [25].

### 3.3. Mutations in Cancer Genes

Korabiowska et al. investigated the expression of DNA mismatch repair genes and growth arrest DNA damage genes in a non-specified cutaneous chromatophoroma in a jumping viper (*Atropoides picadoi*) and cutaneous melanoma in a bullsnake (*Pituophis catenifer sayi*) [35]. They found mutations in the growth arrest DNA damage gene GADD34 and mutations in the DNA mismatch repair gene MLH1 in both cases, and mutations in the DNA mismatch repair gene MSH2 in the melanoma [35]. It is unclear whether these mutations represent potential driver mutations of carcinogenesis.

## 4. Diagnosis

### 4.1. Cytology

Cytologic examination, including fine needle aspirates and impression smears, can be a minimally invasive method to diagnose chromatophoromas. In a few reptile cases this diagnostic technique successfully identified the mass as a chromatophoroma [19,20,21,23,27,36,37]. In all these cases, the diagnosis was subsequently confirmed by histologic examination. Cytologic findings using Romanowsky-type stains included discrete to aggregated stellate, spindloid, or round cells that contained scant to copious amounts of fine, intracytoplasmic dark blue to black granular material [19,20,21,23,27,36,37]. Neoplastic melanophores contained green, black, or brown melanin granules [19,20,21,27,37]. Neoplastic iridophores contained olive green or light brown refractile granules [23,36]. Microscopic examination with polarized light can be utilized to identify iridophoromas, as the granules are birefringent [23,36]. There can be abundant extracellular pigment granules [20,23,27] (Figure 3a,b).

Cytologic examination can be challenging or misleading due to potential aspiration of normal chromatophores in the dermis or macrophages containing phagocytized granulocyte pigment, nuclear debris, or pigment granules from lysed chromatophores in an inflammatory lesion, which can have a similar cytologic appearance to chromatophoromas. Therefore, cytologic cases suspicious for chromatophoroma should always be confirmed by histologic examination.

### 4.2. Histology

Cutaneous chromatophoromas are primarily dermal, but can extend into the subcutis, underlying skeletal muscle, or vertebral column [9,12,13,24,38,39]. Malignant tumors tend to have an infiltrative growth pattern [9,12,13,32,40], but ultimate diagnosis is often based on metastatic spread. In two large case series of chromatophoromas in lizards and snakes (that included 26 and 42 cases, respectively), cases with substantial anaplasia (anisokaryosis and anisocytosis, nuclear atypia, and high mitotic counts) were associated with an increased likelihood of recurrence and/or metastasis in various species [9,12]. It should be noted that histologically benign appearing neoplasms (well-pigmented, mild anisokaryosis and anisocytosis, low mitotic counts) also were found to be metastatic in a few cases; so, presence, but not absence, of lymphatic invasion is a reliable indicator of more aggressive behavior [9,12]. Neoplastic cells are typically spindloid, less commonly epithelioid or polygonal, and with some frequency show a mixture of spindloid and polygonal cells [9,12,13]. Iridophoromas contain fine, light brown, or olive green cytoplasmic pigment that is birefringent under polarized light [9,12,13] (Figure 4a,b and Figure 5a,b). Xanthophoromas often contain red, brown, yellow, or orange cytoplasmic pigment [9,12] (Figure 4c). Melanophoromas contain fine, brown, or black cytoplasmic pigment [9,12,13] (Figure 4d,e and Figure 5c–e). Mixed chromatophoromas are composed of a mixture of the previously described chromatophores [9,12,13]. Mixed melanophoroma–iridophoromas are most common [9,12,13], but an individual case of a mixed iridophoroma–xanthophoroma has been reported in a veiled chameleon (*Chamaeleo calyptratus*) [16]. The degree of pigmentation in chromatophoromas is usually high [9,12,13]. In most tumors, there is at least some degree of pigmentation [9,12,13]. Anisokaryosis and anisocytosis are often mild to moderate [9,12,13]. The mitotic count is typically low (0–2 mitotic figures in 10 high power fields) [9,12,13]. A subset of the bearded dragon (*Pogona* sp.) chromatophoromas and a single melanophoroma in a northern red-bellied cooter (*Pseudemys rubriventris*) were diagnosed as mucinous chromatophoromas based on the presence of abundant Periodic acid-Schiff positive, mucinous stroma within the neoplasm [9,13,19] (Figure 4e). Ulceration of the overlying surface is not uncommon in chromatophoromas (29 out of 208 cases, when reported in the literature) due to their raised cutaneous appearance [12,13,18,22,23,24,36,41,42]. Intraepithelial neoplastic cells are only described in a small number of snake cases, which most likely represent epithelial invasion rather than junctional activity, as chromatophores do not reside in the epidermis in most reptiles [9,24,43]. Lymphatic invasion is uncommonly observed (7 out of 208 cases, when reported in the literature), even in cases that had widespread metastasis [9,12,20,21]. Neoplastic cells at metastatic sites typically have a similar histologic appearance as the primary neoplasm [9,12,17,19,37] (Figure 5a–e).

In reptiles, the term chromatophoroma is applied to a number of pigmented neoplasms, but it is a non-specific in regard to histogenesis and biological behavior. It is therefore important to further characterize the type of chromatophoroma (e.g., melanophoroma, iridophoroma, or xanthophoroma) based on the histologic features of the neoplastic cells when possible. This will help with elucidating potential clinical and behavior differences between the subtypes of chromatophoromas. The modifier “malignant” should be reserved for chromatophoromas (or their subtypes) that have evidence of lymphatic invasion or metastasis.

### 4.3. Immunohistochemistry

Most chromatophoromas have some degree of pigmentation within the mass that allows confirmation of chromatophore origin by histologic examination. Immunohistochemistry (IHC) can be helpful to diagnose amelanotic or non-pigmented chromatophoromas [13,17]. Reported IHC results conflict depending on the study, but overall, antibodies utilized to detect mammalian melanocytes do not appear to have the same sensitivity and specificity for detecting reptilian chromatophoromas compared to their mammalian counterparts [9,12,13,16,17,39,40,41,42,44,45]. MelanA and S-100 have been shown to be most consistently expressed in a number of chromatophoromas in reptiles. S-100 is a sensitive but non-specific marker for melanomas in mammals as it also labels glial cells, neurons, Schwann cells, chondrocytes, adipocytes, and dendritic cells [41,46,47,48]. MelanA is a highly specific marker for melanomas in humans and dogs [37,48,49]. Immunoreactivity for PNL-2 and HMB45 has only been reported for a few cases [12,13,42]. PNL2 is a highly sensitive and specific marker for melanomas in multiple species [46,47,48,50]. In human melanomas, HBM45 has a relatively high specificity, but low sensitivity [49]. In cases of reptile chromatophoromas where IHC was performed, 84 out of 95 cases had immunoreactivity for S-100; 31 out of 90 cases had immunoreactivity for MelanA; 8 out of 60 cases had immunoreactivity for PNL2; and 2 out of 43 cases had immunoreactivity for HMB45 [9,12,13,16,17,39,40,41,42,44,45]. Immunohistochemistry does not appear to be helpful to differentiate the different types of chromatophoromas.

### 4.4. Electron Microscopy

Transmission electron microscopy (TEM) allows for visualization of the pigment structures or platelets to definitively confirm the chromatophore lineage of the neoplasm. For a poorly differentiated case of malignant mixed chromatophoroma (iridophoroma and xanthophoroma) in a veiled chameleon (*Chamaeleo calyptratus*), TEM was used to identify the xanthophore component [16]. Ultrastructurally, melanophores contain dense, dark, round to ellipsoid, cytoplasmic structures (melanosomes) [3,43]. Iridophores contain cytoplasmic clear, stacked, empty cleft-like spaces (reflective purine platelets are lost in processing) [3,43]. Xanthophores can contain concentric lamellated structures with a triple limiting membrane (pteridine-filled pterinosomes) and cytoplasmic droplets (carotenoid-filled lipid droplets) [3,6].

## 5. Chromatophoromas in Lacertilla

### 5.1. Family: Agamidae

Chromatophoromas have been described in the following species: dwarf bearded dragon (*Pogona henrylawsoni*), bearded dragon (*Pogona vitticeps* or *Pogona* sp.), red-barred dragon (*Ctenophorus vadnappa*), and a generic agamid [8,9,11,13,23,40,51,52] (Table 1). A search for “chromatophoromas, melanoma, melanophoromas, iridophoromas, and xanthophoromas” of the archives of Northwest ZooPath revealed chromatophoromas in these additional species: uromastyx (*Uromastyx* sp.), Mali uromastyx (*Uromastyx (dispar) maliensis*), and frilled lizard (*Chlamydosaurus kingii*).

### 5.2. Family: Iguanidae

Chromatophoromas have been described in green iguanas (*Iguana iguana*) [8,36,37,44] (Table 2). A search for “chromatophoromas, melanoma, melanophoromas, iridophoromas, and xanthophoromas” of the archives of Northwest ZooPath revealed no additional affected species. 

### 5.3. Family: Eublepharidae

Chromatophoromas have been described in leopard geckos (*Eublepharis macularius*) [8,9,25,32] (Table 3). A search for “chromatophoromas, melanoma, melanophoromas, iridophoromas, and xanthophoromas” of the archives of Northwest ZooPath revealed no additional affected species.

### 5.4. Family: Chamaeleonidae

Chromatophoromas have been described in the following species: veiled chameleon (*Chamaeleo calyptratus*) and a generic chameleon [8,9,11,16,23,52] (Table 4). A search for “chromatophoromas, melanoma, melanophoromas, iridophoromas, and xanthophoromas” of the archives of Northwest ZooPath revealed no additional affected species.

### 5.5. Family: Helodermatidae

Chromatophoromas have been described in the following species: Rio Fuerte beaded lizard (*Heloderma horridum exasperatum*), Gila monster (*Heloderma suspectum*), beaded lizard (*Heloderma horridum*), and Mexican beaded lizard (*Heloderma horridum*) [10,27,53] (Table 5). A search for “chromatophoromas, melanoma, melanophoromas, iridophoromas, and xanthophoromas” of the archives of Northwest ZooPath revealed no additional affected species.

### 5.6. Family: Varanidae

Chromatophoromas have been described in the following species: ornate monitor (*Varanus ornatus*), Savana monitor (*Varanus exanthematicus*), and two generic monitors [8,9,11,52] (Table 6). A search for “chromatophoromas, melanoma, melanophoromas, iridophoromas, and xanthophoromas” of the archives of Northwest ZooPath revealed chromatophoromas in these additional species: black tree monitor (*Varanus beccarii*), dwarf monitor lizard (*Varanus kingorum*), crocodile monitor (*Varanus salvadorii*), and quince monitor (*Varanus melinus*).

### 5.7. Family: Scincideae

A cutaneous melanoma has been described in a generic skink [11]. A search for “chromatophoromas, melanoma, melanophoromas, iridophoromas, and xanthophoromas” of the archives of Northwest ZooPath revealed chromatophoromas in these additional species: blue tongued skink (*Tiliqua* sp.) and shingleback skink (*Tiliqua rugosa*).

### 5.8. Family: Gekkonidae

An iris melanoma has been described in a generic gecko [11]. A search for “chromatophoromas, melanoma, melanophoromas, iridophoromas, and xanthophoromas” of the archives of Northwest ZooPath revealed chromatophoromas in these additional species: leaf tailed gecko (*Uroplatus phantasticus*) and tokay gecko (*Gekko gecko*).

### 5.9. Families: Diplodacylidae, Corytophanidae, Dactyloidae

No previously reported cases of chromatophoromas were found for *Diplodacylidae*, *Corytophanidae*, or *Dactyloidae*; however, a search for “chromatophoromas, melanoma, melanophoromas, iridophoromas, and xanthophoromas” of the archives of Northwest ZooPath revealed chromatophoromas in these additional species: giant new Caledonian gecko (*Rhacodactylus leachianus*), basilisk (*Basiliscus basiliscus*), and orient knight anole (*Anolis equestris*). The giant new Caledonian gecko was diagnosed with a dermal chromatophoroma with no report of metastasis. The basilisk was diagnosed with a malignant melanoma of the skin with metastasis to the liver and coelom. One orient knight anole was diagnosed with a colonic melanoma with no report of metastasis, and another was diagnosed with an amelanotic melanoma of the soft tissues of the leg with no report of metastasis.

## 6. Chromatophoromas in Serpentes

### 6.1. Family: Colubridae

Chromatophoromas have been described in the following species: pine snake (*Pituophis melanoleucus*), northern pine snake (*Pituophis melanoleucus melanoleucus*), Florida pine snake (everglade snake; *Elaphe obsoleta rossalleni*), gopher snake (bullsnake; *Pituophis catenifer* var. *sayi*), yellow rat snake (*Elaphe obsoleta quadrivittata*), eastern yellowbelly racer (*Coluber constrictor flaviventris*), northern water snake (*Nerodia sipedon sipedon*), Mexican hognose snake (*Heterodon nasicus kennerlyi*), garter snake (*Thamnophis sirtalis terrestrism*), southern water snake (*Nerodia fasciata*), corn snake (*Pantherophis guttatus*), San Francisco garter snake (*Thamnophis sirtalis tetrataenia*), eastern king snake (*Lampropeltis getula*), tricolor hognose snake (*Xenodon pulcher*), California king snake (*Lampropeltis getula californiae*), lyre snake (*Trimorphodon biscutatus*), eastern hognose snake (*Heterodon platirhinos*), Honduran milk snake (*Lampropeltis triangulum hondurensis*), western fox snake (*Pantherophis* spp.), eastern indigo snake (*Drymarchon couperi*), green vine snake (*Ahaetulla nasuta*), rat snake (*Elaphe [Pantherophis] obsolete*), great plains rat snake (*Pantherophis emoryi*), Sierra mountain kingsnake (*Lampropeltis zonata*), and eleven generic colubrids [8,9,10,11,12,15,18,26,35,38,42,43,45,54] (Table 7). A search for “chromatophoromas, melanoma, melanophoromas, iridophoromas, and xanthophoromas” of the archives of Northwest ZooPath revealed chromatophoromas in these additional species: western fox snake (*Pantherophis vulpinus*), common garter snake (*Thamnophis sirtalis*), Greer’s king snake (*Lampropeltis Mexicana greeri*), Andean milk snake (*Lampropeltis triangulum andesiana*), marsh snake (mangrove salt marsh snake; *Nerodia clarkii*), speckled racer (*Drymobius margaritiferus*), narrow-headed garter snake (*Thamnophis rufipunctatus*), eastern plains garter snake (*Thamnophis radix*), Taiwan beauty snake (*Orthriophis taeniuria*), California mountain kingsnake (*Lampropeltis zonata*), western hognose snake (*Heterodon nasicus*), and Chicago garter snake (*Thamnophis sirtalis semifascitus*).

### 6.2. Family: Viperidae

Chromatophoromas have been described in the following species: canebreak (timber) rattlesnake (*Crotalus horridus atricaudatus*), jumping viper (*Atropoides picadoi*), pigmy rattlesnake (*Sistrurus* spp.), bush viper (*Atheris* spp.), northern Pacific rattlesnake (*Crotalus oreganus*), Mojave desert sidewinder (*Crotalus cerastes*), neotropical rattlesnake (*Crotalus durissus*), broad-banded copperhead (*Agkistrodon laticinctus*), temple viper (*Tropidolaemus wagleri*), prairie rattlesnake (*Crotalus viridis*), two generic crotalids, and two generic vipers [8,9,11,12,35,39,42] (Table 8). A search for “chromatophoromas, melanoma, melanophoromas, iridophoromas, and xanthophoromas” of the archives of Northwest ZooPath revealed chromatophoromas in these additional species: Borneal temple viper (*Tropidolaemus wagleri*), broad-banded copperhead (*Agkistrodon laticinctus*), western massasauga (*Sistrurus catenatus tergeminus*), Taylor’s cantil (*Agkistrodon taylori*), Mexican moccasin (*Agkistrodon bilineatus*), black-tailed rattlesnake (*Crotalus molossus*), desert massasauga (*Sistrurus catenatus edwardsii*), fer-de-lance snake (*Bothrops asper*), southern Pacific rattlesnake (*Crotalus oreganus helleri*), and western diamondback rattlesnake (*Crotalus atrox*).

### 6.3. Family: Pythonidae

Chromatophoromas have been described in the following species: reticulated python (*Python reticulatus*), green tree python (*Morelia* (previously *Chondropython*) *viridis*), Burmese python (*Python molurus*), woma python (*Aspidites ramsayi*), and carpet python (*Morelia spilota*) [8,12,22,28] (Table 9). A search for “chromatophoromas, melanoma, melanophoromas, iridophoromas, and xanthophoromas” in the archives of Northwest ZooPath revealed chromatophoromas in this additional species: ball python (*Python regius*).

### 6.4. Family: Boidae

Chromatophoromas have been described in the following species: boa constrictor (*boa constrictor constrictor*), Amazon tree boa (*Corallus hortulana*), yellow anaconda (*Eunectes notaceus*), rainbow boa (*Epicrates cenchria*), anaconda (*Eunectes murinus*), rubber boa (*Charina bottae*), rosy boa (*Charina trivirgata*), Dumeril’s ground boa (*Acrantophis dumerili*), and three generic boids [8,9,11,12,17,55] (Table 10). A search for “chromatophoromas, melanoma, melanophoromas, iridophoromas, and xanthophoromas” in the archives of Northwest ZooPath revealed chromatophoromas in this additional species: Virgin Island boa (*Epcrates monoensis granti*).

### 6.5. Family: Elapidae

Chromatophoromas have been described in the following species: common death adder (*Acanthophis antarcticus*), strap-snouted brown snake (*Pseudonaja aspidorhynchus)*, tiger snake (*Notechis scutatus*), eastern brown snake (*Pseudonaja textilis*), Mengden’s brown snake (*Pseudonaja mengdeni*), black mamba (*Dendroaspis polylepis*), and cape cobra (*Naja nivea*) [12,24,41,55] (Table 11). A search for “chromatophoromas, melanoma, melanophoromas, iridophoromas, and xanthophoromas” in the archives of Northwest ZooPath revealed chromatophoromas in these additional species: brown snake (*Storeria dekayi*) and spitting cobra.

### 6.6. Family: Xenopeltidae

No previously reported cases of chromatophoromas were found for Xenopeltidae; however, a search for “chromatophoromas, melanoma, melanophoromas, iridophoromas, and xanthophoromas” of the archives of Northwest ZooPath revealed a chromatophoroma in a sunbeam snake (*Xenopeltis* sp.). The sunbeam snake was diagnosed as a probable cutaneous amelanotic melanoma with no report of metastasis.

## 7. Chromatophoromas in Chelonia

### 7.1. Family: Testudinidae

Chromatophoromas have been described in the following species: Hermann’s tortoise (*Testudo hermanni*) and red-footed tortoise (*Chelonoidis carbonaria*) [8,9,20,21] (Table 12). A search for “chromatophoromas, melanoma, melanophoromas, iridophoromas, and xanthophoromas” of the archives of Northwest ZooPath revealed no additional affected species.

### 7.2. Family: Emydidae

Chromatophoromas have been described in the following species: northern red-bellied cooter (*Pseudemys rubriventris*) and red-eared slider turtle (*Trachemys scripta elegans*) [19,56] (Table 13). A search for “chromatophoromas, melanoma, melanophoromas, iridophoromas, and xanthophoromas” of the archives of Northwest ZooPath revealed no additional affected species.

## 8. Clinical Staging and Treatment

As in other species, clinical staging is helpful to assess the extent of neoplastic disease, predict prognosis, and form the best therapeutic plan for the patient. Additional clinical tests often include complete blood count (CBC) and blood chemistry, radiographs, ultrasound, computed tomography, or magnetic resonance imaging (MRI) [57]. The results of clinical staging were mentioned only in a small subset of reports of chromatophoromas in reptiles (7 cases out of 208) [21,23,32,36,45]. It was reported to be helpful to access the extent of neoplastic disease in these cases [21,23,32,36,45].

Surgical excision is the most common treatment for chromatophoromas in reptiles [9,12,13]. Complete surgical excision is curative in some cases, but the results of clinical staging can be helpful to determine if surgical excision is likely to be curative [9,12,13,32,36,45]. Complete surgical excision appeared to be curative in 63 out of 208 cases, though it should be noted that loss to follow up or lack of complete postmortem examination may falsely alter these numbers [9,12,13,32,36,45]. Complete surgical excision can be difficult to obtain in some cases due to the infiltrative growth pattern common to reptile chromatophoromas [9,12,13,32,40].

There are a few reported cases of reptile chromatophoromas treated with radiation with variable success. A malignant melanophoroma in a Hermann’s tortoise (*Testudo hermanmi*) was treated with photon standing field irradiation (6 MV); three treatments of 9 Gy once per week using a linear accelerator (total dose: 27 Gy) after incomplete surgical excision [20]. The tortoise’s condition declined 2 weeks after the last radiation treatment [20]. A melanoma in a common death adder (*Acanthophis antarcticus*) was treated with radioactive gold implants, but therapy was reported to be unsuccessful [41]. A melanin-producing malignant chromatophoroma in a yellow rat snake (*Elaphe obsoleta quadrivittata*) was treated with radiation therapy (6 MV electron beam); four equivalent fractions over a 15-day period (total 6000 cGy) after incomplete surgical excision [38]. The scales in the irradiated area became discolored and roughened, but the surgery site healed uneventfully [38]. The snake was irritable for several months and anorexic for 20 weeks following radiation therapy [38]. The snake died 10-months later and necropsy revealed a mass in the coelomic cavity with similar gross features to the previously removed cutaneous chromatophoroma [38]. No histologic examination was performed to confirm the identity of this mass [38].

In mammals, additional treatment options for melanomas include chemotherapy and melanoma vaccines [58]. Unfortunately, little is known about chemotherapy levels, dosages, and effects in reptiles for even the most common neoplasms, so this is still an area that needs investigation [57].

## 9. Conclusions

Chromatophoromas are a relatively common, primarily cutaneous neoplasm in reptiles, with an increasing number of reports in recent years [8,9,12,13]. The increased incidence and apparent species predispositions may be due in part to varying popularity of certain species as pets and exhibit animals, closer monitoring by caretakers, and advancements in veterinary medicine rather than true increased incidence. Some of the larger cases series originated from a small number of pathology groups, so overlapping and therefore over reporting of individual cases and species may have occurred.

Cutaneous chromatophoromas most commonly present as pigmented masses, but for clinicians it is also important to note that they can appear as flat, pigmented or nonpigmented scales [8,9,12,13]. Most chromatophoromas develop spontaneously, but UV (especially artificial UV) radiation may play a role in some cases, especially sun-loving reptiles such as bearded dragons [8]. A genetic component has also been described in an inbred population of leopard geckos [25]. Mutations in tumor suppressor genes most likely play an important role in chromatophoroma oncogenesis in reptiles, just like in mammals [25,35].

Cytologic examination can be helpful to diagnose chromatophoromas [19,20,21,23,27,36,37]. However, histologic examination is required for a definitive diagnosis and allows for assessment of malignant features and margin evaluation. While histologic examination is typically adequate to reach a diagnosis, IHC can be helpful for diagnosing poorly pigmented or amelanotic cases [13,17]. Reports of IHC effectiveness varies among studies and species, emphasizing the need for validation of IHC markers in reptilian species [9,12,13,16,17,39,40,41,42,44,45]. Transmission electron microscopy can be used to definitely identify the chromatophore type based on visualization of pigment structures or platelets. This technique is typically not needed to achieve a definitive diagnosis, but can be used for poorly differentiated chromatophoromas [16]. Regardless, similar to amelanotic melanomas in mammals, ultrastructural identification of early-stage melanosomes can be challenging.

In all reptile families except *Eublepharidae*, melanophore-origin chromatophoromas (melanophoromas) are the most common subtype. In Lacertilia, chromatophoromas are relatively common in the family *Agamidae* (especially bearded dragons) [8,9,11,13,23,40,51,52]. A myxoid variant of the melanophoroma has been described [9,13]. Metastasis is fairly uncommon, but has been reported, with the most commonly affected organs being lungs, livers, kidneys, intestine, and fat bodies [8,9,40]. In the family *Iguanidae* (green iguanas), there have been reports of metastasis [8,36,37,44]. In the family *Eublepharidae* (leopard geckos), malignant iridophoromas are most common with occasional metastasis being described [8,9,25,32]. In the family *Chamaeleonidae* (veiled chameleon), occasional metastases have been described, with the most commonly affected organs being lungs, skin, testicles, spleen, liver, kidneys, heart, gut, fat bodies, and coelomic cavity [8,9,16,23]. In the family *Helodermatidae*, there are no reports of metastasis [10,27,53] and in the family *Varanidae*, rare metastasis have been reported [11].

In Serpentes, chromatophoromas are most common in the family *Colubridae* followed by the family *Viperidae* [9,12]. In the family *Colubridae*, mixed chromatophoromas (melanophoroma and iridophoroma) are also relatively common. All described types of chromatophoromas have a potential for metastasis in this family with the most commonly affected organs being liver, coelomic cavity, spleen, and kidneys [8,9,10,11,12,15,18,26,35,38,42,43,45,54]. In the family Viperidae, metastasis has been relatively commonly reported [8,9,11,12,35,39,42]. In the *Pythonidae* family, metastases are rare [12,22]. In the *Boidae* family, metastasis is relatively frequent [8,9,12,17,55]. In the *Elapidae* family, metastases are rare, with the tissues around the esophagus being the most commonly affected [24,41]. Chromatophoromas have rarely been reported in Chelonians with all reports being melanophoromas and half were metastatic [8,9,19,20,21,56].

The primary therapy is complete surgical excision [9,12,13]. This is curative for some, but given the infiltrative growth pattern and metastatic potential of chromatophoromas in multiple reptile species, surgical excision may not be adequate to control neoplastic spread [9,12,13,32,40]. Cases with substantial anaplasia (anisokaryosis and anisocytosis, nuclear atypia, and high mitotic counts) have been associated with an increased likelihood of recurrence and/or metastasis in some species [9,12]. Absence of lymphatic invasion does not appear to be a reliable indicator of benign behavior [9,12,20,21]. Monitoring and clinical staging is especially important in these cases. Other treatment modalities have not been well studied in reptiles.

## Figures and Tables

**Figure 1 vetsci-09-00115-f001:**
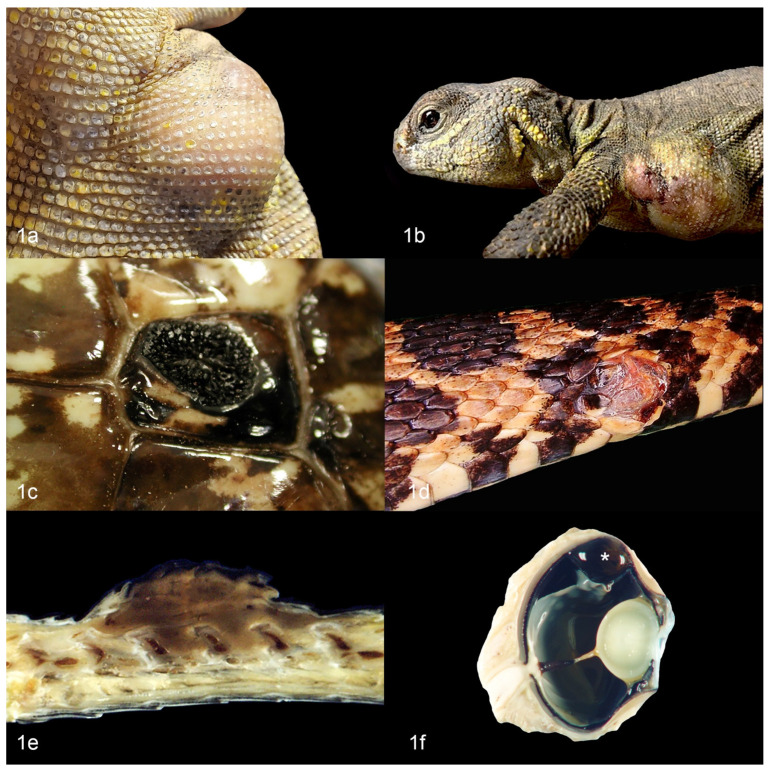
Gross appearance of cutaneous chromatophoromas in reptiles. (**a**) Cutaneous iridophoroma on the axilla of an Uromastyx (*Uromastyx geyri*). (**b**) Cutaneous chromatophoroma on the axilla of an Uromastyx (*Uromastyx geyri*). (**c**) Cutaneous melanoma on the body wall of a cape cobra (*Naja nivea*). (**d**) Cutaneous melanoma on the body wall of a western fox snake (*Pantherophis* spp.). (**e**) Cross section of cutaneous melanoma on the dorsum of a rat snake (*Elaphe [Pantherophis] obsolete*). The tumor extends into the underlying connective tissues between the dorsal spinal facets. (**f**) Uveal melanoma (asterisk) from a bearded dragon (*Pogona* sp).

**Figure 2 vetsci-09-00115-f002:**
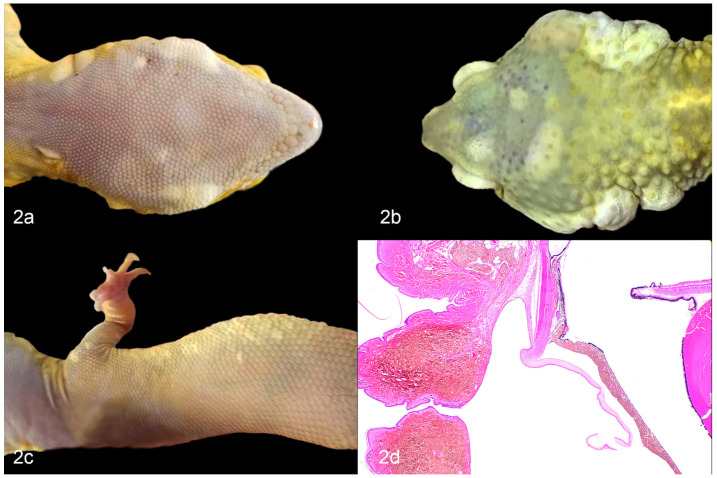
Iridophoromas in ‘lemon frost’ leopard geckos (*Eublepharis macularius*). (**a**) Multiple white, nodular, cutaneous iridophoromas along the ventral mandible. (**b**) Multiple white, smooth to nodular, cutaneous iridophoromas on the dorsal aspect of the head and bilaterally within the periaural skin. (**c**) Multiple, pale yellow, smooth, pigmented scales of a cutaneous iridophoroma. (**d**) Iridophoroma in the iris, conjunctiva, periocular skin, and connective tissues. H&E.

**Figure 3 vetsci-09-00115-f003:**
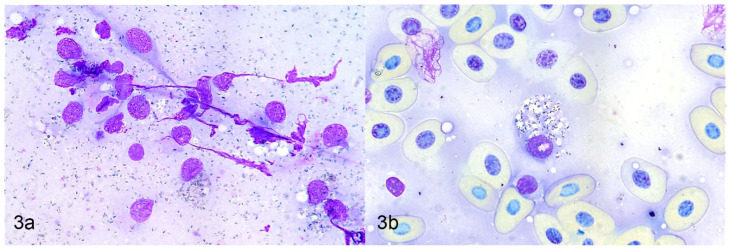
Cytologic aspirates of a chromatophoroma in a canebreak (timber) rattlesnake (*Crotalus horridus atricaudatus*). Romanowsky-type stain. (**a**) Few individualized spindloid cells with intracytoplasmic blue-black pigmented granules and numerous identical extracellular granules. (**b**) Round cell with numerous cytoplasmic pigment granules.

**Figure 4 vetsci-09-00115-f004:**
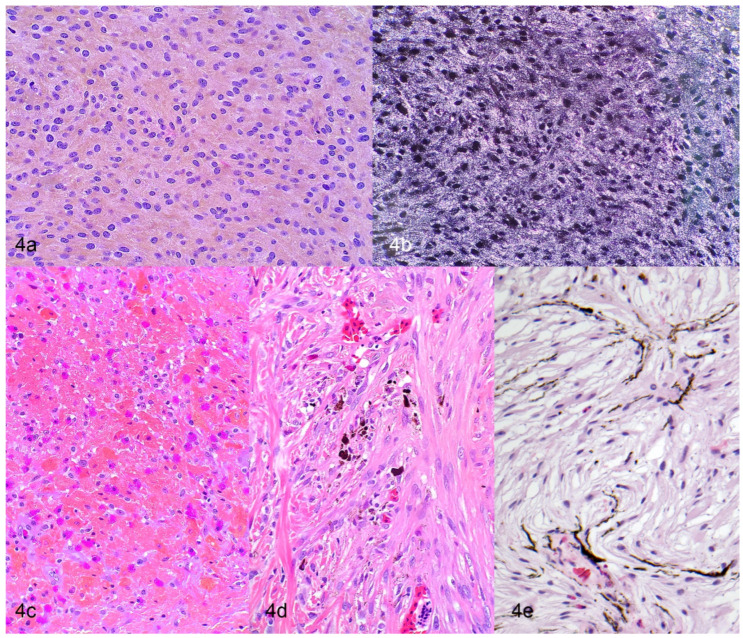
Histologic features of cutaneous chromatophoromas in reptiles. (**a**,**b**) Iridophoroma in a tricolor hognose snake (*Xenodon pulcher*). H&E (**a**) and H&E with polarized light (**b**). (**c**) Xanthophoroma in a corn snake (*Pantherophis guttatus*). Note the deep orange intracellular pigment. H&E. (**d**) Poorly differentiated spindle cell variant of amelanotic melanoma in a corn snake (*Pantherophis guttatus*). H&E. (**e**) Mucinous melanoma in a bearded dragon (*Pogona* sp). H&E.

**Figure 5 vetsci-09-00115-f005:**
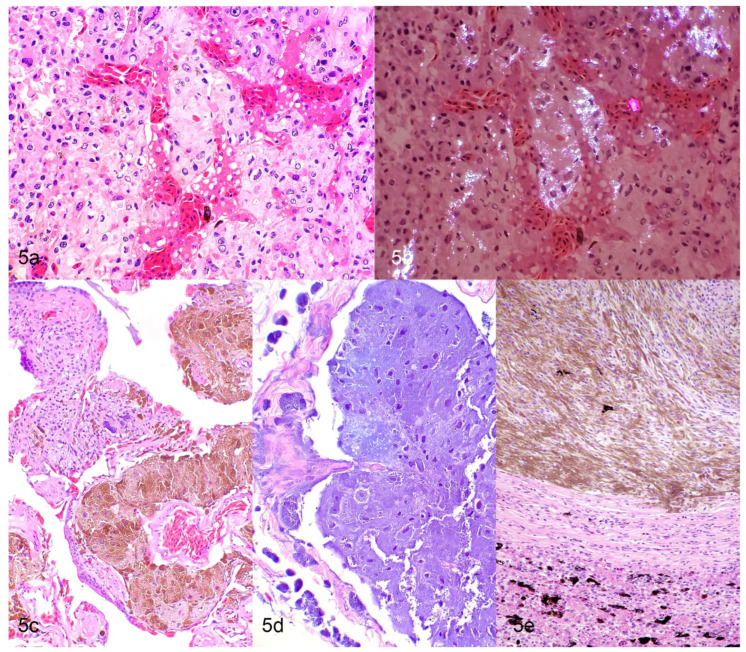
Histologic features of metastatic chromatophoromas in reptiles. (**a**,**b**) Hepatic metastasis of an iridophoroma in a veiled chameleon (*Chamaeleo calyptratus*). H&E (**a**) and H&E with polarized light (**b**). (**c**,**d**) Pulmonary metastasis of a melanoma in a rosy boa (*Charina trivirgata*). H&E (**c**) and H&E with bleach (**d**). (**e**) Hepatic metastasis of a melanoma in a chameleon (*Chamaeleo* sp.). H&E.

**Table 1 vetsci-09-00115-t001:** Clinical features of chromatophoromas in the reptile family: *Agamidae*.

Species	Site	Diagnosis	Presence of Visceral Metastases	Organs Affected by Metastases	Citation
Dwarf bearded dragon (*Pogona henrylawsoni*)	Cutaneous and oral	Malignant iridophoroma	Yes	Heart, liver, kidney, intestine, pharyngeal and sublingual mucosa, visceral fat	[40]
Red-barred dragon (*Ctenophorus vadnappa*)	Cutaneous	Malignant iridophoroma	Yes; tumor microembolic in pulmonary arterial vessels	Lung	[23]
Bearded Dragon (*Pogona* sp).	Cutaneous or lip	Melanoma (4 cases)	No		[8,9,13,51,52]
		Melanophoroma (26)	Yes, 2 cases	Kidneys, liver, lungs, intestine, fat bodies	
		Myxoid melanophoroma (24)	No		
		Iridophoroma (8)	Yes, 1 case	11 different (unspecified) organs	
		Mixed chromatophoroma (melanophoroma and iridophoroma) (5)	No		
		Nonpigmented chromatophoroma (6)	No		
Generic Agamid	Cutaneous	Melanoma	Not reported		[11]

**Table 2 vetsci-09-00115-t002:** Clinical features of chromatophoromas in the reptile family: *Iguanidae*.

Species	Site	Diagnosis	Presence of Visceral Metastases	Citation
Green iguana (*Iguana iguana*)	Cutaneous, nasal, or unspecified	Iridophoroma (1 case)	No	[8,36,37,44]
		Melanophoroma (3)	No	

**Table 3 vetsci-09-00115-t003:** Clinical features of chromatophoromas in the reptile family: *Eublepharidae*.

Species	Site	Diagnosis	Presence of Visceral Metastases	Organs Affected by Metastases	Citation
Leopard gecko (*Eublepharis macularius*)	Cutaneous	Malignant iridophoroma (4 cases)	Yes, 1 case; No, 3 cases	Liver, eye, muscle	[8,9,25,32]
		Melanophoroma (1)	Yes	Lung	

**Table 4 vetsci-09-00115-t004:** Clinical features of chromatophoromas in the reptile family: *Chamaeleonidae*.

Species	Site	Diagnosis	Presence of Visceral Metastases	Organs Affected by Metastases	Citation
Veiled chameleon (*Chamaeleo calyptratus*)	Cutaneous	Malignant mixed chromatophoroma (iridophoroma and xanthophoroma) (1 case)	Yes	Skin, lung, testicle	[8,9,16,23,52]
		Metastatic iridophoroma (1)	Yes	Skin, liver, spleen, kidney, abdominal fat, lung, stomach, peritoneum, omentum, myocardium, epicardium, great vessels, testicle, skeletal muscle	
		Melanoma (1)	Not reported		
		Melanophoroma (3)	Yes; 1 case	Heart, lung, tongue, stomach, gut, liver, spleen, kidneys, bone, fat bodies, parietal serosa of the coelomic cavity, brain	
		Iridophoroma (3)	No		
		Erythro-/xanthophoroma (1)	No		
		Mixed tumors (melanophoroma and iridophoroma) (1)	No		
Generic chameleon	Cutaneous	Melanoma	Not reported		[11]

**Table 5 vetsci-09-00115-t005:** Clinical features of chromatophoromas in the reptile family: *Helodermatidae*.

Species	Site	Diagnosis	Presence of Visceral Metastases	Citation
Rio Fuerte beaded lizard (*Heloderma horridum exasperatum*)	Pulmonary	Melanophoroma	No	[27]
Gila monster (*Heloderma suspectum*)	Not specified	Melanoma	Not reported	[53]
Beaded lizard (*Heloderma horridum*)	Body wall	Melanoma	Not reported	[53]
Mexican beaded lizard (*Heloderma horridum*)	Cutaneous	Chromatophoroma	No	[10]

**Table 6 vetsci-09-00115-t006:** Clinical features of chromatophoromas in the reptile family: *Varanidae*.

Species	Site	Diagnosis	Presence of Visceral Metastases	Organs Affected by Metastases	Citation
Ornate monitor (*Varanus ornatus*)	Cutaneous	Malignant melanoma	Not reported		[52]
Savanna monitor (*Varanus exanthematicus*)	Cutaneous	Iridophoroma	Not reported		[8,9]
Generic monitors	Cutaneous	Melanoma (2 cases)	Yes (1 case)	Not reported	[11]

**Table 7 vetsci-09-00115-t007:** Clinical features of chromatophoromas in the reptile family: *Colubridae*.

Species	Site	Diagnosis	Presence of Visceral Metastases	Organs Affected by Metastases	Citation
Pine snake (*Pituophis melanoleucus*)	Cutaneous or lip	Malignant melanoma (1 case)	Yes	Cloaca, coelomic cavity, liver	[10,18]
		Melanosarcoma (1)	No		
		Malignant chromatophoroma (1)	Not reported		
Northern pine snake (*Pituophis melanoleucus melanoleucus*)	Cutaneous	Combined iridophoroma and melanophoroma	Yes	Thyroid gland, subepicardium, liver, spleen, lung, kidneys	[43]
Florida pine snake (everglade snake; *Elaphe obsoleta rossalleni*)	Cutaneous	Malignant melanoma	Yes	Skeletal muscle, bone marrow of vertebral bodies, pulp of maxillary teeth, dental ridges, lung, fat body, liver, spleen, kidneys, testes, hemipenes, pancreas	[54]
Gopher snake (bullsnake; *Pituophis catenifer* var. *sayi*)	Cutaneous or unspecified	Malignant chromatophoroma (mosaic, xanthophore and melanophores) (1)	No		[10,12,15,26,35,42]
		Melanoma (2)	Yes, 1 case; other not reported	Liver	
		Chromatophoroma (1)	Not reported		
		Mixed chromatophoroma (melanophoroma and iridophoroma) (2)	No		
		Iridophoroma (1)	No		
Yellow rat snake (*Elaphe obsoleta quadrivittata*)	Cutaneous	Melanin-producing malignant chromatophoroma (1)	Yes	Coelomic cavity	[12,38]
		Melanocytoma (1)	No		
Eastern yellowbelly racer (*Coluber constrictor flaviventris*)	Cutaneous	Chromatophoroma	No		[45]
Northern water snake (*Nerodia sipedon sipedon*)	Cutaneous	Chromatophoroma and benign dermal melanoma	Not reported		[10]
Mexican hognose snake (*Heterodon nasicus kennerlyi*)	Cutaneous	Chromatophoroma	Not reported		[10]
Garter snake (*Thamnophis sirtalis terrestrism*)	Cutaneous or unspecified	Malignant mixed chromatophoroma (1)	Not reported		[8,9,15]
		Melanophoroma (2)	Yes, 1 case; No, 1 case	Not reported	
		Erythro-/ Xanthophoroma (4)	Yes in 4 cases	Widespread: specific organs not reported	
Southern water snake (*Nerodia fasciata*)	Unspecified	Melanophoroma	No		[8,9]
Corn snake (*Pantherophis guttatus*)	Cutaneous	Malignant mixed chromatophoroma (melanophoroma and iridiophoroma) (1)	Yes	Not reported	[12]
		Iridophoroma (1)	No		
San Francisco garter snake (*Thamnophis sirtalis tetrataenia*)	Cutaneous	Mixed chromatophoroma (melanophoroma and iridiophoroma) (2)	No		[12]
		Melanocytoma (1)	No		
		Iridophoroma (1)	No		
Eastern king snake (*Lampropeltis getula*)	Cutaneous	Mixed chromatophoroma (melanophoroma and iridiophoroma)	No		[12]
Tricolor hognose snake (*Xenodon pulcher*)	Cutaneous	Mixed chromatophoroma (melanophoroma and iridiophoroma) (1)	No		[12]
		Malignant iridophoroma (1)	Yes	Not reported	
California king snake (*Lampropeltis getula californiae*)	Cutaneous	Mixed chromatophoroma (melanophoroma and iridiophoroma)	No		[12]
Lyre snake (*Trimorphodon biscutatus*)	Cutaneous	Melanocytoma	No		[12]
Eastern hognose snake (*Heterodon platirhinos*)	Cutaneous	Melanocytoma	No		[12]
Honduran milk snake (*Lampropeltis triangulum hondurensis*)	Cutaneous	Melanocytoma	No		[12]
Western fox snake (*Pantherophis* spp.)	Cutaneous	Melanocytoma	No		[12]
Eastern indigo snake (*Drymarchon couperi*)	Cutaneous	Melanocytoma	No		[12]
Green vine snake (*Ahaetulla nasuta*)	Cutaneous	Melanocytoma	No		[12]
Rat snake (*Elaphe [Pantherophis] obsolete*)	Cutaneous	Melanocytoma	No		[12]
Great plains rat snake (*Pantherophis emoryi*)	Cutaneous	Malignant melanoma	Yes	Not reported	[12]
Sierra mountain kingsnake (*Lampropeltis zonata*)	Cutaneous	Malignant melanoma	Yes	Not reported	[12]
Generic colubrids	Cutaneous	Melanoma (11)	No		[11]

**Table 8 vetsci-09-00115-t008:** Clinical features of chromatophoromas in the reptile family: *Viperidae*.

Species	Site	Diagnosis	Presence of Visceral Metastases	Organs Affected by Metastases	Citation
Canebreak rattlesnake (*Crotalus horridus atricaudatus*)	Cutaneous	Malignant chromatophoroma	Not reported		[39]
Jumping viper (*Atropoides picadoi*)	Cutaneous	Chromatophoroma	Not reported		[35,42]
Pigmy rattlesnake (*Sistrurus* spp.)	Not specified	Melanophoroma	No		[8,9]
Bush viper (*Atheris* spp.)	Cutaneous	Xanthophoroma (1 case)	No		[12]
		Iridophoroma (1)	No		
Northern pacific rattlesnake (*Crotalus oreganus*)	Cutaneous	Melanocytoma	No		[12]
Mojave Desert sidewinder (*Crotalus cerastes*)	Cutaneous	Melanocytoma (1)	No		[12]
		Malignant melanoma (1)	Yes	Not reported	
Neotropical rattlesnake (*Crotalus durissus*)	Cutaneous	Melanocytoma	No		[12]
Broad-banded copperhead (*Agkistrodon laticinctus*)	Cutaneous	Melanocytoma	No		[12]
Temple viper (*Tropidolaemus wagleri*)	Cutaneous	Malignant melanoma	Yes	Not reported	[12]
Prairie rattlesnake (*Crotalus viridis*)	Cutaneous	Malignant melanoma	Yes	Not reported	[12]
Generic vipers and crotalids	Cutaneous	Melanoma (4)	Yes (1 case)	Not reported	[11]

**Table 9 vetsci-09-00115-t009:** Clinical features of chromatophoromas in the reptile family: *Pythonidae*.

Species	Site	Diagnosis	Presence of Visceral Metastases	Organs Affected by Metastases	Citation
Reticulated python (*Python reticulatus*))	Cutaneous	Melanoma (1 case)	Yes	Coelomic cavity, kidneys	[22]
		Multiple non-malignant melanomas (1)	No		
Green tree python (*Morelia* (previously *Chondropython*) *viridis*)	Small intestine; cutaneous	Chromatophoroma (1)	Not reported		[12,28]
		Malignant iridophoroma (1)	Yes	Not reported	
Burmese python (*Python molurus*)	Unspecified	Melanophoroma	Not reported		[8]
Woma python (*Aspidites ramsayi*)	Cutaneous	Melanocytoma	No		[12]
Carpet python (*Morelia spilota*)	Cutaneous	Melanocytoma	No		[12]

**Table 10 vetsci-09-00115-t010:** Clinical features of chromatophoromas in the reptile family: *Boidae*.

Species	Site	Diagnosis	Presence of Visceral Metastases	Organs Affected by Metastases	Citation
Boa constrictor (*boa constrictor constrictor*)	Oral; unspecified	Metastatic malignant amelanotic melanoma (1 case)	Yes	Liver, spleen	[8,9,17]
		Melanophoroma (1)	No		
Amazon tree boa (*Corallus hortulana*)	Cutaneous	Malignant melanophoroma	Vascular dissemination	Not reported	[55]
Yellow anaconda (*Eunectes notaceus*)	Cutaneous	Melanophoroma	Yes	Heart, stomach, gut, pancreas, kidneys, uterus, ovaries, fat body	[8,9]
Rainbow boa (*Epicrates cenchria*)	Cutaneous	Melanocytoma	No		[12]
Anaconda (*Eunectes murinus*)	Cutaneous	Melanocytoma	No		[12]
Rubber boa (*Charina bottae*)	Cutaneous	Melanocytoma	No		[12]
Rosy boa (*Charina trivirgata*)	Cutaneous	Melanocytoma (1)	No		[12]
		Malignant melanoma (1)	Yes	Not reported	
Dumeril’s ground boa (*Acrantophis dumerili*)	Cutaneous	Malignant melanoma	Yes	Not reported	[12]
Generic boids	Cutaneous	Melanoma (3)	No		[11]

**Table 11 vetsci-09-00115-t011:** Clinical features of chromatophoromas in the reptile family: *Elapidae*.

Species	Site	Diagnosis	Presence of Visceral Metastases	Organs Affected by Metastases	Citation
Common death adder (*Acanthophis antarcticus*)	Cutaneous	Melanoma	Yes	Adventitia of esophagus, pericardium	[41]
Strap-snouted brown snake (*Pseudonaja aspidorhynchus)*	Cutaneous	Malignant melanophoroma	Yes	Esophagus, intestine, liver, descending aorta	[24]
Tiger snake (*Notechis scutatus*)	Cutaneous	Malignant melanophoroma	No		[24]
Eastern brown snake (*Pseudonaja textilis*)	Cutaneous	Melanophoroma	No		[24]
Mengden’s brown snake (*Pseudonaja mengdeni*)	Cutaneous	Malignant melanophoroma	No		[24]
Black mamba (*Dendroaspis polylepis*)	Disseminated, specific sites unspecified	Malignant melanoma	Not reported		[55]
Cape cobra (*Naja nivea*)	Cutaneous	Mixed chromatophoroma (melanophoroma and iridophoroma)	No		[12]

**Table 12 vetsci-09-00115-t012:** Clinical features of chromatophoromas in the reptile family: *Testudinidae*.

Species	Site	Diagnosis	Presence of Visceral Metastases	Organs Affected by Metastases	Citation
Hermann’s tortoise (*Testudo hermanni*)	Cutaneous (carapace)	Malignant melanophoroma	Yes	Lungs, liver, spleen, kidneys, adrenal glands, thyroid gland, ductus deferens	[8,9,20]
Red-footed tortoise (*Chelonoidis carbonaria*)	Cutaneous	Melanoma	No		[21]

**Table 13 vetsci-09-00115-t013:** Clinical features of chromatophoromas in the reptile family: *Emydidae*.

Species	Site	Diagnosis	Presence of Visceral Metastases	Organs Affected by Metastases	Citation
Northern red-bellied cooter (*Pseudemys rubriventris*)	Cutaneous	Mucinous variant of melanophoroma	Not reported		[19]
Red-eared slider turtle (*Trachemys scripta elegans*)	Cutaneous (shell)	Metastatic melanophoroma	Yes	Not reported	[56]

## Data Availability

Not applicable.

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
