# Peer review of "Chromatophoromas in Reptiles"

_vetsci, 2022, doi:10.3390/vetsci9030115_

Round 1

Reviewer 1 Report

The authors provide a review on chromatophoromas in reptiles. This work can be of interest to pathologists and clinicians who work with these animals as these tumors are relatively common.

The text is described in a comprehensive manner and the figures are of a very good quality.

There are a few things that should be reviewed before publication.

TEXT:

Lines 183, 243, 246 – Replace Romanowki by Romanowsky.

I understand that it may not be the aim of this specific study but it would be nice to have more details on the criteria used to diagnose benign vs malignant tumors based on histology. In the text the terms used are melanophoromas, melanomas, melanocytomas, melanosarcomas, and malignant melanomas. I understand that this is a review and the authors of the reviewed papers named these tumors in different ways but I think it would be nice if you could include more information on how to name these properly based on the features found.

Lines 221-223 – Here you state that the presence of intraepithelial neoplastic cells most likely represents an epithelial invasion as chromatophores do not reside in epithelia but at the beginning of the manuscript (lines 31-32) you mention that few reptiles can have epidermal melanocytes. This seems to be contradictory. Maybe you could say “since chromatophores do not reside in the epidermis in most reptiles” or something similar.

Line 252, table 9, line 497 – The following data appear in several parts of the manuscript: 2/4/2022 8:49:00 AM. Please delete it.

Line 276 – Replace “pterinsomes” by “pterinosomes”.

Line 277 – Add a full stop after the references.

Line 287 – The full stop is in line 288.

Line 438 – As there are not published data for this family it would be interesting to include where the tumor was located in the sunbeam snake.

FIGURES AND TABLES:

The figures are not mentioned in the text following an order. The first figure mentioned is figure 6, then figures 1-5, figures 7-10, figures 20-21, figures 11-2, figure 15 and figure 16. Please review the figure citation and order recommendations of the journal.

I cannot see the reference of figures 13 and 14 in the text.

I would recommend to indicate where the tumor is in figure 6 (maybe with an arrow, an asterisk or something similar).

It would be nice to have a scale bar or the magnification in the histology and cytology figures.

With regards to tables, I understand that the number in brackets after the diagnosis is the number of tumors found reported on each species. It may be clearer if you explain it at least the first time you use it.

Regarding the presence of visceral metastasis it may not be necessary to include again the name of each tumor if you make subdivisions of the diagnosis and just mention the presence/absence of metastasis. 

REFERENCES:

Please review the journal recommendations for references.

Some of them may be grouped in the text as 39-42 or 46-48 for example.

Line 256 references 42 and 44 are mentioned twice.

Line 509 reference 42 is mentioned twice.

Reviewer 2 Report

This is an excellent review of chromatophoromas in reptiles. I only have few suggestions to try to improve the information presented to clinician.

Line 15. Abstract: Replace “cases” with “neoplasm” or “tumor” or similar.

Line 56. Replace “confirmation” with “conformation”.

Line 215.  I suspect the information is not available but, if available, please indicate the field size (2.37 square millimeters?). If not leave as it is.

Line 219 “Ulceration of the overlying surface is not uncommon in chromatophoromas due to their raised cutaneous appearance [12][13][18][22][23][26][36][41][42].”   And line 223-224 “Lymphatic invasion 223 is uncommonly observed, even in cases that had widespread metastasis [9][12][20][21]. “

It would be interesting to add how many where ulcerated (or had lymphatic invasion) rather than using “not uncommon” or “uncommonly observed”. Something like X out of X reports of chromatophoromas in the literature indicate that the overlying surface was ulcerated (or lymphatic invasion).

Line 243 nd 246. Replace  “Romanowki” with “Romanowsky”

Line 252 Correct “[13]2/4/2022 8:49:00 AM[17]”

Line 256 to 265. The manuscript would be improved by developing a little more the section on IHC.

“MelanA and S-100 have been shown to be most consistently expressed in a number of chromatophoromas in reptiles” If possible, add how many chromatophoromas in the literature were reported to express these markers and how many did not?

PNL-2 and HMB45 are mentioned but it is not clearly stated how these markers performed in reptiles. What are the results in the few cases where these have been used?

Line 267. Adding EM images to the manuscript would be a nice addition if possible.

Line 330 to 349. Please indicate if visceral metastases have been found in these 3 families. I assume not as nothing is mentioned but it would be best to state it clearly.

Line 403 to 405. “A search for “chromatophoromas, melanoma, 403 melanophoromas, iridophoromas, and xanthophoromas” in the archives of Northwest 404 ZooPath revealed chromatophoromas in these additional species: ball python”. Please, go through the manuscript and change “these” to “this” as appropriate when only one additional species is listed.

Line 440 to 440. It is not indicated that a search in the ZooPath database was performed. I suspect it is an omission and a search was done. Please add that a search did not reveal additional cases.

General comment for section 5, 6 and 7. It would be a nice addition to indicate how often metastases occurred in the reported cases in each family. In which organs the metastases occurred. If there are any organs more commonly affected in each order or family as appropriate. In the summary paragraph lines 516 to 535, the words “occasional”, “uncommon”, “relatively common” are used but it would be good to have the data included in sections 5 to 7 so the reader has an idea if 3 cases or 60 cases were reported for example.  

Line 461-463 “Complete surgical excision is curative in some cases, but the results of clinical staging can be helpful to determine if surgical excision is likely to be curative [9][12][13][32][36][45].”  It would be helpful to clinicians to expend a little more. In cases in the literature when the information is available, how often was surgical excision curative? How often was staging helpful?  Some of this information is indicated in line 539 to 543 but not presented here..

Line 497. Correct “8][9][12][13]2/4/2022 8:49:00 AM.”

Line 539- 541 “Cases with substantial anaplasia (anisokaryosis and anisocytosis, nuclear atypia, and high mitotic counts) have been associated with an increased likelihood of recurrence and/or metastasis in some species 541 [9][12].”  This info is in the conclusion but does not appear in the histology section. Please include it in the appropriate section and discuss briefly. For example, is this association based on 2-3 cases? Which species?
